# Hamiltonian Additional Damping Control for Suppressing Power Oscillation Induced by Draft Tube Pressure Fluctuation

**Yun Zeng [1], Shige Yu [1,\*] , Fang Dao [2], Xiang Li [2], Yiting Xu [2] and Jing Qian [2]**

1   Faculty of Civil Engineering and Mechanics, Kunming University of Science and Technology, Kunming 650500, China; zengyun001@kust.edu.cn
2   Faculty of Metallurgical and Energy Engineering, Kunming University of Science and Technology, Kunming 650093, China
\*   Correspondence: yushige@stu.kust.edu.cn; Tel.: +86-183-8846-5441

**Abstract:** The power oscillation induced by pressure fluctuation in the draft tube of the hydraulic turbine is one of the limiting factors preventing the Francis turbine from operating in the vibration zone. At the present power grid with a high proportion of renewable energy resources, we try to improve the load regulation ability of the hydropower units by extending the stable operation zone to the vibration zone. By the mathematical modelling of pressure fluctuation, this paper gives an analytical expression of the power oscillation. We derive the extended Hamiltonian model of the hydropower unit where power oscillation is external excitation. Secondly, the damping injection method introduces some desired interconnection and damping matrices as the Hamiltonian damping factor into the additional damping control. Finally, through theoretical analysis and experimental simulation, this research discusses the resonance characteristics of pressure fluctuation and power oscillation, the equivalent analysis between the damping factor and equivalent damping coefficient, and the control design of vibration zone crossing during the start-up. Simulation results show that when $r_{25} = 1.3$, the minimum power oscillation amplitude is 0.5466, which is equivalent to an increase in $D$ by 20. The maximum oscillation amplitude decreases by 4.6%, and the operation limited zone is reduced by 10.1%. The proposed additional damping control can effectively suppress the power oscillation and expand the regulation range.

**Keywords:** pressure fluctuation; power oscillation; Hamiltonian damping factor; additional damping control; vibration zone crossing

## 1. Introduction

In a power system with a high proportion of renewable energy resources, hydropower units (HU) as a large-capacity regulating power supply have been the only feasible technical path to date. So, the load regulation requires a high frequency and a large span. When the power oscillation occurs in the deviation from the rated-load condition, the stable operation zone of HU is limited to 60%–100% of the rated load at present, and some are even limited to 70–100% [1]. This situation narrows the range of active power regulation, which cannot meet the requirements for regulating power supply. To expand the regulation range, related research has even investigated the stability of operation under non-design conditions and the expansion of the overload area [2,3]. From the perspective of practical projects, it has greater potential for extending the stable operation zone under part-load conditions. There are some theoretical and experimental studies on the formation and evolution mechanism [4,5] and suppression measures [6,7] of pressure fluctuation in DL. However, because of its complex dynamic behaviour and multi-field coupling characteristics, we have no suitable method to solve this problem completely. This paper only analyses and studies the power oscillation of the active power caused by the pressure fluctuation of DL in the Francis turbine.

When the frequency of pressure fluctuation in DL is close to the other oscillation frequencies of HU, the resonance of hydraulic, mechanical, or electrical systems will be

induced, which can cause greater harm to the entire system. For example, when the frequency is close to the natural frequency of the generator, the power oscillation of active power will be amplified [8–10], and when the frequency is close to the low-frequency oscillation frequency of the power grid, it will even threaten grid stability [11]. The problem of power oscillation caused by pressure fluctuation in hydropower stations has been reported from time to time. There are two main ways: one is that the units operate in the vibration zone under part-load conditions [12]; the next is that the units take too long to cross or stay in the vibration zone during the start-up [13].

In the studies discussing the influence of pressure fluctuation on operation control, its external features or measurable characteristics are usually used to describe pressure fluctuation in DL, which many scholars have explored [14–16]. However, due to the differences in structural parameters and operating conditions of HU, it is difficult to quantify the nonlinear dynamics behaviour of pressure fluctuation accurately. In the related applications of control analysis, the frequency and amplitude of pressure fluctuation are regarded as measurable characteristics. For example, according to the features of the vortex belt in DL under different load conditions, the piecewise function is used to give the variation of pressure fluctuation [17]. The simplest way is to express the amplitude and frequency of pressure fluctuation is by sine or cosine [18]. This paper adopts this simple sine function to describe pressure fluctuation in the experimental simulation.

From the dynamic of view, it is effective to increase the system damping to suppress the parameter variety and disturbance. For example, by increasing the damping of HU, the amplitude of power oscillation can be significantly reduced [19]. The article [20] used PSS to suppress the power oscillation induced by pressure fluctuation, which is equal to increasing the damping of their resonance zone. Based on this idea, this paper attempts to study how to improve the damping of HU to suppress the amplitude of power oscillation from the control.

The damping characteristics are improved by narrowing the oscillation amplitude and reducing the oscillation times through the control strategy. For the power oscillation induced by the vortex belt of DL, the optimization of governor control parameters has a less inhibiting effect on the power oscillation [21]. Therefore, it is necessary to consider other auxiliary signals as control inputs to supply additional damping to the object system, such as the PSS module on the generator side. There are a few control theories whose purpose is to improve the damping characteristics of the system. The damping injection method based on generalized Hamiltonian theory explicitly proposes injecting damping into the system to improve the transient characteristics [22–24]. The core of this theory is to change the geometric structure by modifying the interconnection or damping matrix. However, it is difficult to obtain the analytical expression of the modified Hamilton function to maintain the mathematical equivalence. Some research results have used flexible methods to design the equivalent control law after structural modification in practical problems. For example, for high-order systems with incomplete control, the control design adopts the approximate treatment at the given equilibrium point [25], and the article [26] has proposed a parametric design method called H-damping-assignable. The Hamiltonian function is minimized to synthesize the corresponding control laws [27].

This paper proposes an equivalent control design method of damping injection in the fifth-order Hamiltonian system of HU, which aims at suppressing the power oscillation induced by pressure fluctuation. Associations among the value of damping factor, the variation of damping characteristics, and the influence of resonance characteristics are studied. In practical application, selecting appropriate damping factors and adopting PID + Hamiltonian additional damping control can effectively improve the vibration zone crossing during the start-up of HU. The new control strategy ensures that the system quickly passes through the resonance region and reduces the active oscillation amplitude induced by pressure fluctuation.

## 2. Problem Description

The pressure fluctuation of a Francis turbine results from the upward transmission of an eccentric vortex in the draft tube (DL). It is presented as a periodic change of the water head across DL [18]. The mechanical power $p_t$ defined by the IEEE Working Group [28] is expressed as

$$p_t = A_t h_t (q - q_{nl}) \tag{1}$$

$p_t$ only applies to the unconstrained draft tube with zero inlet pressure, and $h_t$ only refers to the pressure at the hydro turbine admission. However, the strict definition of the water head is the pressure difference between the upstream of the hydro turbine and the downstream.

The new water head accounting for the pressure fluctuation is described by

$$h_t = h_{t0} + h_p \sin(\omega_p t) \tag{2}$$

Noted that $h_{t0}$ is $h_t$ in Equation (1).

Similarly, we define the new mechanical power after inserting Equation (2) into Equation (1):

$$\begin{aligned} p_t &= A_t h_{t0}(q - q_{nl}) + A_t(q - q_{nl})h_p \sin(\omega_p t) \\ &= p_{t0} + \Delta p \end{aligned} \tag{3}$$

where $p_{t0}$ is $p_t$ in Equation (1) and $\Delta p = A_t h_p(q - q_{nl})\sin(\omega_p t)$.

Let $\Delta p / p_{t0}$ in Equation (3), we have:

$$\frac{\Delta p}{p_{t0}} = \frac{h_p}{h_{t0}} \sin(\omega_p t) \tag{4}$$

Equation (4) shows that the frequency of power oscillation is the same as that of pressure fluctuation. According to "Basic technical specification for large and medium hydraulic turbines" [29], the peak-to-peak allowable value is 3~10% of the corresponding operating head, and the absolute value is less than 10 (m). In medium and low turbines, their power oscillation may have a larger oscillation amplitude.

The motion equation of HU is:

$$\frac{d\omega}{dt} = \frac{1}{T_j}[p_t - p_e - D(\omega - 1)] \tag{5}$$

Substituting Equation (3) into Equation (5) to obtain:

$$\frac{d\omega}{dt} = \frac{1}{T_j}[p_{t0} - p_e - D(\omega - 1)] + \frac{1}{T_j}F_p \sin(\omega_p t) \tag{6}$$

We define $F_p = A_t h_p(q - q_{nl})$.

According to the theory of vibration mechanics [30], the active power under harmonic excitation is sinusoidal oscillation with the same external excitation frequency. When the frequency of pressure fluctuation is close to the natural frequency of power oscillation, the latter amplitude will be increased by the resonance, and HU will be unable to operate stably. According to the resonance characteristics, the oscillation amplitude can be reduced or suppressed by increasing the damping of the system. In this paper, we try to minimize the power oscillation and improve the operating stability by the damping injection method.

## 3. Additional Damping Control for the Hamiltonian System

The port-controlled Hamiltonian (PCH) system takes the form of:

$$\dot{x} = [J(x) - R(x)]\frac{\partial H}{\partial x}(x) + g(x)u(x) \tag{7}$$

The state variable $x \in R^n$ and control input $u \in R^m$, $m < n$, $H(x)$ is the energy function, $g(x)$ is the input matrix, the structure matrix $J(x)$ is antisymmetric, and the damping matrix $R(x)$ is a semi-positive definite symmetric matrix.

(i)     Assume that $x_*$ is the equilibrium, the constant control $v_*$ is the solution of

$$0 = [J(x_*) - R(x_*)]\frac{\partial H}{\partial x}(x_*) + g(x_*)v_* \tag{8}$$

If the corresponding conditions [31] are satisfied, $v_*$ is the input of stabilization control in system (7).

(ii)     Given the constant structural modification $J_a$ and $R_a$, the desired Hamiltonian structure matrix is $J_d(x) = J(x) + J_a$, $R_d(x) = R(x) + R_a$, which satisfies $J_d(x) = -J_d(x)^{\mathrm{T}}$, $R_d(x) = R_d(x)^{\mathrm{T}}$. So, the system (7) can be rewritten as

$$
\begin{aligned}
\dot{x} &= [(J_d(x) - J_a) - (R_d(x) - R_a)]\frac{\partial H}{\partial x}(x) + g(x)u(x) \\
&= [J_d(x) - R_d(x)]\frac{\partial H}{\partial x}(x) - [J_a - R_a]\frac{\partial H}{\partial x}(x) + g(x)u(x) \\
&= [J_d(x) - R_d(x)]\frac{\partial H}{\partial x}(x) + g(x)\beta(x)
\end{aligned}
\tag{9}
$$

where

$$g(x)\beta(x) = -[J_a - R_a]\frac{\partial H}{\partial x}(x) + g(x)u(x) \tag{10}$$

$\beta(x)$ is the equivalent control after structure modification.

At the equilibria $x_*$, the modified Hamiltonian system (9) also satisfies:

$$
\begin{aligned}
0 &= [J_d(x_*) - R_d(x_*)]\frac{\partial H}{\partial x}(x_*) + g(x_*)\beta(x_*) \\
&= [J(x_*) - R(x_*)]\frac{\partial H}{\partial x}(x_*) + [J_a - R_a]\frac{\partial H}{\partial x}(x_*) + g(x_*)\beta(x_*) \\
&= -g(x_*)v_* + [J_a - R_a]\frac{\partial H}{\partial x}(x_*) + g(x_*)\beta(x_*)
\end{aligned}
\tag{11}
$$

When the system is asymptotically stable, the control $\beta(x)$ in Equation (10) replacing $\beta(x_*)$ in a neighbourhood around the equilibrium point is valid. Assume that $g(x)$ is a constant matrix, that is, $g(x) = g(x_*) = g$. Combining with Equations (10) and (11), there are:

$$g[u(x) - v_*] = (J_a - R_a)[\frac{\partial H}{\partial x}(x) - \frac{\partial H}{\partial x}(x_*)] \tag{12}$$

(iii)     Let $\alpha(x) = u(x)-v_*$. If $g(x)$ is full of rank, $(g^{\mathrm{T}}g)$ is reversible. In addition,

$$\alpha(x) = (g^T g)^{-1}g^T(J_a - R_a)[\frac{\partial H}{\partial x}(x) - \frac{\partial H}{\partial x}(x_*)] \tag{13}$$

$\alpha(x)$ is the additional damping control of the Hamiltonian system (7) with $(J_a, R_a)$, and $\alpha(x_*) = 0$. Notice that $\alpha(x)$ only plays a role in the transient process.

The resulting control law must calculate the matching Equation (8) at the given equilibrium $x_*$, which is its limitation. The control design method is selected by the load state of the system in the actual operation. The additional damping control is appropriate for this research, which aims to suppress the power oscillation induced by pressure fluctuation.

## 4. Hamiltonian Model of Hydropower Units

As the external excitation, the power oscillation induced by the pressure fluctuation in DL is added to the motion Equation (5) of HU. The generalized Hamiltonian model is from the transformation of the differential equation model. Assuming that the power oscillation approximates the Hamiltonian system input as an external excitation, we add the excitation to the PCH model of HU [32]. The expanded PCH model is expressed as follows:

$$\dot{x} = [J(x) - R(x)]\frac{\partial H}{\partial x} + g(x)v(x) + F(x) \tag{14}$$

$$J(x) = \begin{bmatrix} 0 & C_T(x) & 0 & 0 & 0 \\ -C_T(x) & 0 & 0 & \frac{1}{2T_jT_y} & 0 \\ 0 & 0 & 0 & \frac{1}{T_j} & 0 \\ 0 & -\frac{1}{2T_jT_y} & -\frac{1}{T_j} & 0 & 0 \\ 0 & 0 & 0 & 0 & 0 \end{bmatrix},$$

$$R(x) = \begin{bmatrix} r(x) & 0 & 0 & 0 & 0 \\ 0 & r(x) & 0 & \frac{1}{2T_jT_y} & 0 \\ 0 & 0 & 0 & 0 & 0 \\ 0 & \frac{1}{2T_jT_y} & 0 & \frac{D}{T_j^2\omega_B} & 0 \\ 0 & 0 & 0 & 0 & \frac{\omega_B X_{ad}^2}{T'_{d0}X_f} \end{bmatrix}$$

$$g(x) = \begin{bmatrix} 0 & \frac{1}{T_y} & 0 & 0 & 0 \\ 0 & 0 & 0 & 0 & \frac{\omega_B}{T'_{d0}} \end{bmatrix}^T, v(x) = \begin{bmatrix} u_p(x) \\ E_f \end{bmatrix}$$

$$F(x) = \begin{bmatrix} 0 & 0 & 0 & \frac{1}{T_j}F_p\sin(\omega_p t) & 0 \end{bmatrix}^T$$

where $x = [x_1, x_2, x_3, x_4, x_5]^T = [q, y, \delta, \omega, E'_q]^T$, $F(x)$ is the external excitation, and the Hamilton function is:

$$H(x) =$$
$$T_y A_t \frac{x_1^2}{x_2}(x_1 - q_{nl}) + \frac{1}{2}T_j\omega_B x_4^2 - \frac{1}{2}U_s^2\left(\frac{1}{X_{q\Sigma}} - \frac{1}{X'_{q\Sigma}}\right)\cos^2 x_3 \tag{15}$$
$$+ \frac{1}{2}\frac{U_s^2}{X_{q\Sigma}} + \frac{1}{2}\frac{X_{d\Sigma}X_f}{X_{ad}^2 X'_{d\Sigma}}x_5^2 - \frac{U_s\cos x_3}{X'_{d\Sigma}}x_5$$

The algebraic equations in Equation (14) are:

$$\begin{cases} f_1(x) = \frac{1}{T_w}\left(h_0 - f_p x_1^2 - \frac{x_1^2}{x_2^2}\right) \\ f_2(x) = -\frac{1}{T_y}(x_2 - y_0) \\ r(x) = \frac{-f_2(x)\nabla_{x2}H + A_t x_1^3/x_2}{(\nabla_{x1}H)^2 + (\nabla_{x2}H)^2} \\ C_T(x) = \frac{f_1(x) + \nabla_{x1}Hr(x)}{\nabla_{x2}H} \\ u_p(x) = u(x) + T_y\frac{f_1(x)\nabla_{x1}H + A_t x_1^3/x_2}{\nabla_{x2}H} \\ p_t = A_t\frac{x_1^2}{x_2^2}(x_1 - q_{nl}) \end{cases}$$

where $u_p(x)$ is the control realized by a feedback dissipative, $X_{d\Sigma} = X_d + X_T + X_L$, $X'_{d\Sigma} = X'_d + X_T + X_L$ and $X_{q\Sigma} = X_q + X_T + X_L$.

In the above model, the hydro turbine is a second-order model including flow and guide vane opening. The head losses are proportional to flow squared and given by $f_1(x)$ in $C_T(x)$ of structure matrix $J(x)$ in Equation (14). It is only a simple hydraulic condition with a penstock and a non-elastic water column. If the elastic water column is adopted, the model becomes a five-order model, and the derived Hamiltonian model has a more complex expression, while HU is the eighth-order model. It is bound to make additional damping control (13) more difficult.

In this paper, we first design the additional damping control for the fifth-order Hamiltonian model. Then, the adaptability of the control law for the expanded PCH system is studied by example simulation.

## 5. Control Design

### 5.1. Damping Injection

This research aims to inject damping into the Hamiltonian system and suppress the power oscillation. In the PCH system (7), the damping matrix $R(x)$ reflects the port dissipation characteristics. We want to add the corresponding Hamiltonian damping factor

$R_a$ to $R(x)$ to increase the system damping. In HU, the active power belongs to the electrical systems, and the pressure fluctuation of DL belongs to the hydraulic systems. So, $R_a$ is the correlation item between hydraulic and electrical systems. The desired structure matrix is selected as follows:

$$J_a = \mathbf{0}_{5\times5}, \ \ \boldsymbol{R}_a = \begin{bmatrix} 0 & 0 & 0 & 0 & 0 \\ 0 & 0 & 0 & 0 & r_{25} \\ 0 & 0 & 0 & 0 & 0 \\ 0 & 0 & 0 & 0 & 0 \\ 0 & r_{25} & 0 & 0 & 0 \end{bmatrix} \tag{16}$$

Insert (14) and (16) into (13), and the additional damping control is as follows:

$$\boldsymbol{\alpha}(x) = \begin{bmatrix} \alpha_1(x) \\ \alpha_2(x) \end{bmatrix} = \begin{bmatrix} T_y r_{25}\left[\frac{\partial H}{\partial x_5}(x) - \frac{\partial H}{\partial x_5}(x_*)\right] \\ \frac{T'_{d0}}{\omega_B} r_{25}\left[\frac{\partial H}{\partial x_2}(x) - \frac{\partial H}{\partial x_2}(x_*)\right] \end{bmatrix} \tag{17}$$

Combining $u(x) = \boldsymbol{\alpha}(x) + v_*$ to obtain:

$$\begin{aligned}
&u(x) \\
&= \begin{bmatrix} u_{p*} + T_y r_{25}\left[\frac{X_{d\Sigma} X_f}{X_{ad}^2 X'_{d\Sigma}}(x_5 - x_{5*}) - \frac{U_s}{X'_{d\Sigma}}(\cos x_3 - \cos x_{3*})\right] \\ E_{f*} - \frac{T'_{d0}}{\omega_B} r_{25} T_y A_t\left[\frac{x_1^2}{x_2^2}(x_1 - q_{nl}) - \frac{x_{1*}^2}{x_{2*}^2}(x_{1*} - q_{nl})\right] \end{bmatrix}
\end{aligned} \tag{18}$$

$u(x)$ is a state feedback control, which builds the coupling correlation between the different systems.

According to Lyapunov stability theory, the range of $r_{25}$ in (16) should be calculated by the positive definiteness of the Hessian matrix. Only in this way is the PCH system with the desired structure matrix $(\boldsymbol{J}_d, \boldsymbol{R}_d)$ asymptotically stable. However, for a high-order system, it is too complicated to confirm the positive definite condition of the Hessian matrix, and the obtained range of $r_{25}$ at the given equilibrium is not the expected optimal value. Therefore, this paper adopts the numerical example simulation to determine its value range.

*5.2. Analysis of Control Law*

The control input $u_p(x)$ in the PCH system (14) can be rewritten as:

$$u_p(x) = u(x) + \frac{-\frac{1}{T_w} T_y A_t \Delta h \frac{1}{x_2}(3x_1^2 - 2x_1 q_{nl}) + A_t x_1^3/x_2}{-T_y p_t} \tag{19}$$

where

$$\Delta h = -\left(h_0 - f_p x_1^2 - \frac{x_1^2}{x_2^2}\right) \tag{20}$$

The change in water head $\Delta h$ differs in the calculation between the rigid and elastic water-hammer models, and the turbine power $p_t$ and flow $q$ are both affected by $\Delta h$. From the above discussion, it seems feasible to calculate $\Delta h$ in the elastic water hammer model and add it to the additional damping control of the PCH system. This problem needs more research by simulation.

*5.3. Control Structure*

The control input of the hydro turbine is the main servomotor displacement $u$, and that of the generator is excitation voltage $E_f$. We can regard the additional damping control (17) as a part of coordinated control consisting of the traditional speed governor and excitation controller. The structure of the Hamiltonian additional damping control is shown in Figure 1.

In the structure diagram, we give the port information of the existing control system and the scheme of additional control signal access. The speed governor adopts parallel PID control: $K_p = 5.0$, $K_I = 1.7$, $K_D = 1.3$, and the excitation controller adopts thyristor

excitation PID: $K_{P1} = 1.0$, $K_{I1} = 1.5$, $K_{D1} = 0.0001$. The main parameters of the system are $A_t = 1.127$, $T_w = 2.242$, $T_y = 0.5$, $T_j = 8.999$, $T_{d0'} = 5.4$, $X_d = 1.07$, $X_d' = 0.34$, $X_q = 0.66$, $X_f = 1.29$, $X_{ad} = 0.97$, $D = 5$.

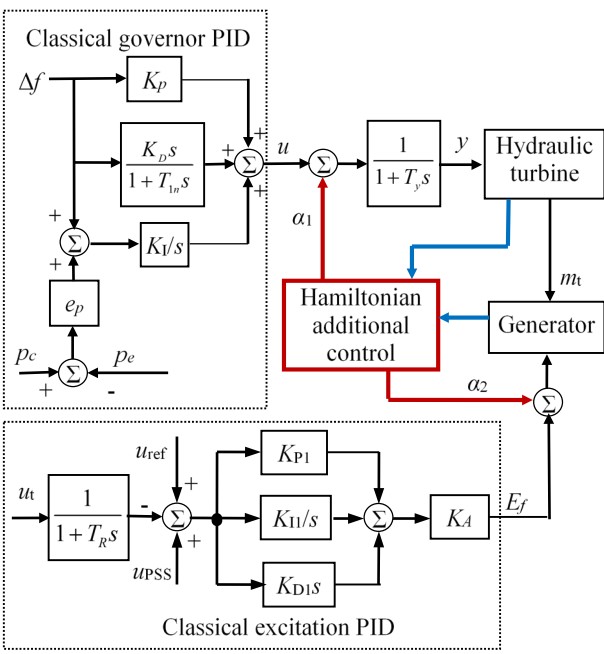

**Figure 1.** Control structure.

## 6. Simulation Research

### 6.1. Pressure Fluctuation Analysis

The frequency of pressure fluctuation is usually described by multiples of that of rated rotation in HU. The approximate estimation at present generally is to delimit an adequate range. The natural frequency is nearly 1.0 (Hz). We define $f_p$ as the pressure fluctuation frequency and $f_p = \omega_p / 2\pi$. In the following simulation, we try to find the possible resonance phenomenon with $f_p = 0.9$ (Hz), $h_p = 0.05$ (p.u) and $f_p = 0.7$ (Hz), $h_p = 0.05$ (p.u). The pressure fluctuation occurs at $t = 1.0$ (s). The mechanical power $p_t$ of the hydro turbine and the active power $p_e$ of the generator is shown in Figure 2.

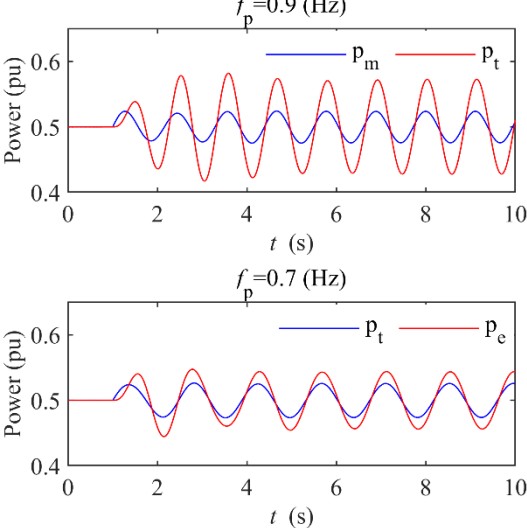

**Figure 2.** $p_t$ and $p_e$ with different $f_p$ s.

$p_t$ and $p_e$ have the same oscillation frequency after entering the steady state. When $f_p = 0.9$ (Hz) is close to the natural frequency (1.0 (Hz)), there is a larger difference in the oscillation amplitude between $p_e$ and $p_t$ than that of $f_p = 0.6$ (Hz). Here is a resonance zone of the pressure fluctuation and power oscillation. The article [18] also found that the frequency of pressure fluctuation has a great influence on the oscillation amplitude of active power, and there is an amplification point of the maximum amplitude. This result is consistent with the classical vibration mechanics theory.

According to the classical theory of vibration mechanics, increasing the damping can suppress the amplification of oscillation amplitude in the resonance zone, which is the theoretical basis of this paper.

### 6.2. Damping Injection Simulation

Let $h_p = 0.05$ (p.u), $f_p = 0.9$ (Hz), and the pressure fluctuation occurs at $t = 1.0$ (s). The responses of $p_e$ under the control laws (17) with different $r_{25}$s are shown in Figure 3.

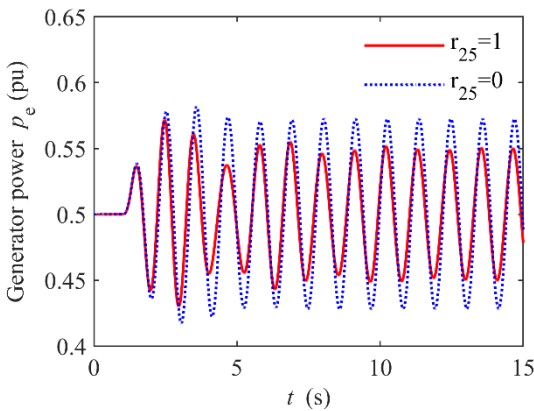

**Figure 3.** Comparison of power oscillations with different $r_{25}$s.

$r_{25} = 0$ is equivalent to not adding the additional damping in control (17), and $r_{25} = 1$ is added to the damping. Compared to the result of no control, the Hamiltonian additional damping control can obviously narrow the amplitude of power oscillation.

As mentioned in Section 4, the value of $r_{25}$ needs to be determined by simulation. $r_{25}$ has a reasonable range to guarantee the convergence of the system to the desired neighbourhood. The oscillation amplitude changes with $r_{25}$ are as follows in Figure 4.

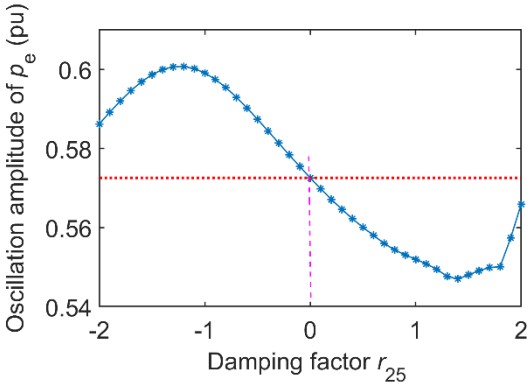

**Figure 4.** Amplitude change of $p_e$ with $r_{25}$.

When $r_{25} = 0$, the oscillation amplitude of $p_e$ is the reference value marked by a red dotted line.

When the data points are above the reference line, $\alpha(x)$ with $r_{25} < 0$ supplies negative damping. In addition, $\alpha(x)$ with $r_{25} > 0$ shows a positive damping characteristic. This

variation of damping characteristics is consistent with the expectation of the damping injection method. When $r_{25}$ exceeds the range limited to the X-axis in Figure 4, the response curve of $p_e$ will gradually diverge and lose its stability, and when $r_{25} = 1.3$, the oscillation amplitude is the smallest, which can function as the optimal value.

The above study determines the effective range and optimal value of the damping factor through simulation calculation. In this simple way, we solve the problem of manually constructing and solving the high-order Hessian matrix.

Further simulation shows that this change in damping characteristics is related to the frequency of pressure fluctuation in DL. Given different $r_{25}$s, the oscillation amplitude change with $f_p$ is shown in Figure 5.

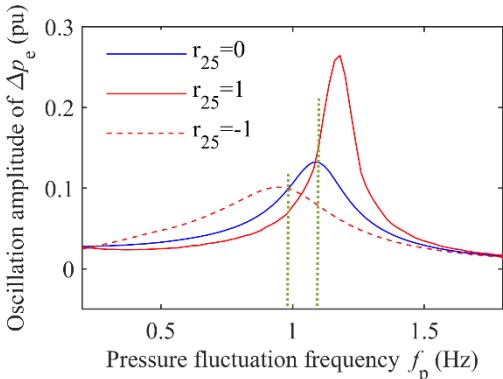

**Figure 5.** Amplitude change with $f_p$ in different $r_{25}$s.

The oscillation amplitude with $r_{25} = 0$ is also used as the reference value marked by the blue line. The intersection point of any two lines tells that the damping characteristic changes dynamically with $f_p$. When $f_p$ is below a specific value located at the intersection, the positive damping induced by $r_{25} = 1$ adjusts to the negative damping. This situation also applies to $r_{25} = -1$.

From the two waveform peaks in Figure 5, the resonance peak point is shifted with the value of $r_{25}$. The additional damping control based on the desired structure matrix ($J_d$, $R_d$) can modify the self-vibration characteristics and natural frequency. This phenomenon reveals a new way to improve the oscillation characteristic of the system, which needs further study.

*6.3. Damping Injection Quantization*

It is difficult to make rigorous theoretical calculations to quantify the damping in the Hamiltonian addition control. The power oscillation caused by the equivalent damping coefficient of generator $D$ can be equated with that of a certain $r_{25}$. We try to obtain the value of $r_{25}$ indirectly by calculating $D$.

Take $r_{25} = 0$, $D = 25$ and $r_{25} = 1.3$, $D = 5$ for example, we select the maximum amplitude of power oscillation in steady state as a reference. The responses under two conditions are shown in Figure 6.

The oscillation amplitude of the red line is 0.5477, which is close to that of the blue line (= 0.5466) in $t = 10 \sim 15$ s. It shows that the effect of $r_{25} = 1.3$ is equivalent to an increase in $D$ by 20.

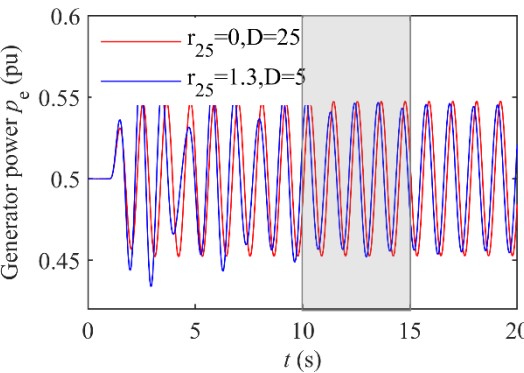

**Figure 6.** Damping equivalence calculation.

### 6.4. Expand the Stable Operation Zone

Figure 4 shows that the Hamiltonian additional damping control with any $r_{25}$s cannot completely restrain power oscillation induced by pressure fluctuation. The expansion of the oscillation amplitude is the main cause of the loss of stability. We want to use the Hamiltonian additional damping control to expand the load regulation range, which is analysed as follows with an example simulation.

Under different load conditions, traditional PID control and PID + Hamiltonian additional damping control are adopted. The comparison is as follows in Figure 7.

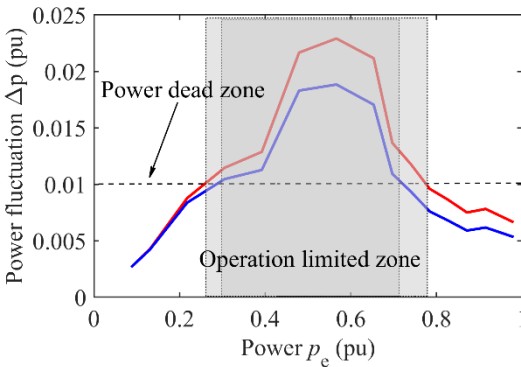

**Figure 7.** Operation limited zone with different controls.

The dead-zone set in Figure 8 is 0.01 (p.u). When Hamiltonian additional damping control is added, the power oscillation (blue line) is lower than the PID control (red line). In HU, the operation limited zone (shaded part) shrinks, and the load regulation range is extended.

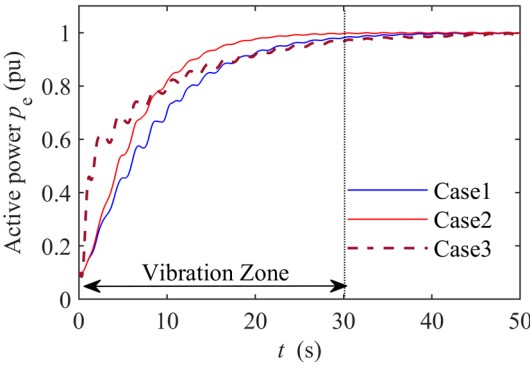

**Figure 8.** Response curves of $p_e$ with different controls.



*6.5. Simulation of Vibration Zone Crossing*

From no-load to rated load, HU will inevitably pass through the part-load vibration zone. If it takes too long in this process, the power oscillation in the vibration zone will induce the low-frequency oscillation [18].

When the active power increases from 0.1 to 1.0 (p.u), the unit will cross the vibration zone during the part-load operation. Consider the following three conditions.

Case 1. No additional Hamiltonian additional damping control, that is, $r_{25} = 0$, only the PID control of speed regulation and excitation.

Case 2. Add Hamiltonian additional damping control, and $r_{25} = -0.5$. In the transient process of load increase, the target value in control law is obtained by the difference between the initial and final active values of $p_e$.

Case 3. Add Hamiltonian additional damping control, and $r_{25} = -0.5$. The target value only corresponds to load $p_e = 1$, ignoring its variation in the load increase.

The simulation results are shown in Figure 8.

From the above figure, $p_e$ of different control strategies receives different degrees of influence from pressure pulsation when the unit runs in the part-load vibration zone during the start-up (0~30 s). After 30 s, the unit leaves the vibration zone and gradually enters a steady state, and the influence of pressure pulsation disappears. The power oscillation may occur when it crosses the vibration zone too slowly or when the pressure fluctuation amplitude is high. Compared to the control effect of Case 1, the control law in Case 2 can greatly reduce the amplitude of power oscillation, and Case 3 can let the unit pass through the vibration zone rapidly.

From the signal flowchart in Figure 1, the Hamiltonian additional damping control is directly added to the PID output. It is equivalent to adding a power feed-forward to improve the regulation speed. The opening speed of the guide vane is accelerated, which may cause high pressure in the pipeline. During the start-up, we should comprehensively consider the relationship between the velocity of passing through and the pressure change.

## 7. Conclusions

In this paper, we first give the calculation formula of the water head considering the pressure fluctuation in DL and expand the PCH model of HU. Secondly, the damping injection method is used to design the Hamiltonian additional damping control to suppress the power oscillation induced by pressure fluctuation. There are three conclusions:

1.  The simulation results show that adding the Hamiltonian damping factor is mathematically equivalent to increasing the oscillation damping, and it is effective to use additional damping control based on the damping injection method.
2.  The resonance point of pressure fluctuation and power oscillation shifts with the values of the Hamiltonian damping factor. The damping characteristic of the same factor is a variation of positive-negative near the resonance point. In application, the values of these damping factors should be selected by the load condition.
3.  PID + Hamiltonian additional damping control can expand the stable operation zone. During the start-up, HU applying the Hamiltonian additional damping control can faster pass through the vibration zone and have a smaller power oscillation than the PID control.

However, in deriving the control strategy, the approximate treatment at the given equilibrium point is used to obtain the equivalent control law. This processing method is only a flexible approach selected for practical problems. It is difficult to maintain mathematical equivalence between the modified and original systems, and due to the excessively strict mathematical equivalence, there are still many difficulties in applying it to high-order systems. These issues will continue to be explored in future research.

**Author Contributions:** J.Q., F.D., X.L. and Y.X. has contributed the most to the conception, guidance, and revising of the manuscript. The data acquisition and analysis were completed by Y.Z. The

manuscript was written by S.Y. All authors have read and agreed to the published version of the manuscript.

**Funding:** This research was funded by National Natural Science Foundation of China, grant number 52079059.

**Data Availability Statement:** Data sharing is not applicable to this article as no datasets were generated or analysed during the current study.

**Acknowledgments:** Not applicable.

**Conflicts of Interest:** The authors declare no conflict of interest.

## Nomenclature

| | |
|---|---|
| DF | Draft tube |
| HU | Hydropower unit |
| PCH | Port-controlled Hamiltonian |
| $A_t$ | Constant proportionality factor |
| $h_t$ | Water head of the hydro turbine (p.u) |
| $q$ | Flow of the hydro turbine (p.u) |
| $q_{nl}$ | No-load flow of the hydro turbine (p.u) |
| $h_p$ | Amplitude of Pressure fluctuation in DL (p.u) |
| $\Delta p$ | Mechanical power oscillation (p.u) |
| $T_j$ | Inertia time constant (s) |
| $p_e$ | Active power of the generator (p.u) |
| $D$ | The equivalent damping coefficient |
| $Fp$ | Amplitude of power oscillation in the hydro turbine (p.u) |
| $u$ | Speed governor output (p.u) |
| $E_f$ | Excitation controller output (p.u) |
| $y$ | Guide vane opening (p.u) |
| $y_0$ | Initial guide vane opening (p.u) |
| $T_w$ | Water inertia time (s) |
| $f_p$ | Water head loss coefficient |
| $T_y$ | Time constant of the main servomotor (s) |
| $E_{q'}$ | Internal transient voltage (p.u) |
| $U_s$ | Infinite-bus voltage (p.u) |
| $X_d$ | The d-axis synchronous reactance |
| $X_d'$ | Transient reactance of the generator (p.u) |
| $X_T$ | Reactance of the transformer (p.u) |
| $X_L$ | Reactance of the transmission line (p.u) |
| $X_q$ | The q-axis synchronous reactance (p.u) |
| $X_f$ | Reactance of the excitation winding (p.u) |
| $X_{ad}$ | The d-axis armature reaction reactance (p.u) |
| $T'_{d0}$ | Time constant (s) |
| $\omega_p$ | Angular frequency of pressure fluctuation (rad/s) |
| $\omega$ | Angular velocity of the generator (p.u) |
| $\omega_B$ | Basic angular velocity, $\omega B = 314$ rad/s |
| $\delta$ | Power angle (rad) |

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
