# Peer review of "Hamiltonian Additional Damping Control for Suppressing Power Oscillation Induced by Draft Tube Pressure Fluctuation"

_water, doi:10.3390/w15081479_

Round 1

Reviewer 1 Report

Generally paper is good.

I would suggest to increase the discussion at the end of the paper and to increase the sizes' of the figures. 

The idea and logic of the paper is seen, so I would recommend publication after minor changes mentioned above

Author Response

We feel great thanks for your professional review work on our article. As you are concerned, there are several problems that need to be addressed. According to your nice suggestions, we have made extensive corrections to our previous draft, the detailed corrections are listed below.

C1. Generally paper is good.

Thank you for your recognition and encouragement.

C2. I would suggest to increase the discussion at the end of the paper and to increase the sizes' of the figures.

We sincerely appreciate the valuable comments. We have re-written this part (Line 372-377) according to the Reviewer’s suggestion. We discuss the advantages and limitations of this present study and give the need of the future scope. And increase the word count of the article to 4200 words.

C3. The idea and logic of the paper is seen, so I would recommend publication after minor changes mentioned above

Thank you for your reminder.

We tried our best to improve the manuscript and made some changes marked in red in revised paper which will not influence the content and framework of the paper. We appreciate for Reviewers’ warm work earnestly, and hope the correction will meet with approval. Once again, thank you very much for your comments and suggestions.

Reviewer 2 Report

Dear Authors:

The paper discusses a method to suppress power oscillation in Francis turbines caused by pressure fluctuations in the draft tube. The authors develop a mathematical model to analyze the power oscillation and derive an extended Hamiltonian model of the hydro-power unit. They introduce a damping injection method using desired interconnection and damping matrices as the Hamiltonian damping factor into the additional damping control. The authors also discuss the resonance characteristics of pressure fluctuation and power oscillation, equivalent analysis between the damping factor and equivalent damping coefficient, and control design of vibration zone crossing during the start-up. The simulation results show that the proposed additional damping control is effective in suppressing power oscillation induced by pressure fluctuation. Here are some comments to be addressed.

1) The numerical results of the proposed method's performance should be listed in the manuscript's abstract.

2) The authors need to provide comments on the cited papers to explain why their proposed approach is more convincing. This is the authors' contribution, and it helps readers understand the reasoning behind the proposed approach. They should also provide a thorough critical literature review, highlighting the drawbacks of existing approaches and defining the main research direction, methodologies, and unsolved problems. The authors should explain why their proposed approach is suitable to solve the critical problem and provide clear indications of the state-of-the-art development through convincing literature reviews.

3) The methodology of the approach has to be more clearly presented.

4) Comparison with other existing methods is needed to show the benefit of the proposed approach.

5) More numerical simulations are needed to show the performance of the proposed approach.

6) The limitations and complexity of the proposed approach should discussed

Author Response

Dear Reviewers:

On behalf of all the contributing authors, I would like to express our sincere appreciations of your letter and reviewers’ constructive comments concerning our article entitled “Hamiltonian Additional Damping Control for Suppressing Power Oscillation Induced by Draft Tube Pressure Fluctuation” (Manuscript No: water-2327237). These comments are all valuable and helpful for improving our article. According to the reviewers’ comments, we have made extensive modifications to our manuscript and supplemented extra data to make our results convincing. In this revised version, changes to our manuscript were all highlighted within the document by using blue-colored text. Point-by-point responses to the reviewers are listed below this letter.

1) The numerical results of the proposed method's performance should be listed in the manuscript's abstract.

Thank you for your reminder. We have re-written this part(Line 22-26) according to the Reviewer’s suggestion. Provide a numerical quantification of the simulation results.

2) The authors need to provide comments on the cited papers to explain why their proposed approach is more convincing. This is the authors' contribution, and it helps readers understand the reasoning behind the proposed approach. They should also provide a thorough critical literature review, highlighting the drawbacks of existing approaches and defining the main research direction, methodologies, and unsolved problems. The authors should explain why their proposed approach is suitable to solve the critical problem and provide clear indications of the state-of-the-art development through convincing literature reviews.

We sincerely appreciate the valuable comments. We have checked the literature carefully and adjusted reference citations [7,14,23,24]. We have rewritten this part according to the Reviewer’s suggestion (Line 37-40), and references [4,5] have been provided to explain the generation of pressure fluctuations. At the end of the INTRODUCTION part, we have added this part about the application of this study (Line 96-100).

3) The methodology of the approach has to be more clearly presented.

We sincerely appreciate the valuable comments. We have added a reference citation [18], which expresses it by sine or cosine that the amplitude and frequency of pressure fluctuation (Line 103). And We have added a reference citation [32], the vibration mechanics textbook (Line 129).

4) Comparison with other existing methods is needed to show the benefit of the proposed approach.

Thanks for your careful checks. We are sorry for our carelessness. Based on your comments, we have checked the literature carefully and adjusted the reference citation [23,24]. We have rewritten this INTRODUCTION part to explain the necessity of the control strategy adopted in this article(Line 75-91).

5) More numerical simulations are needed to show the performance of the proposed approach.

We think this is an omission we made. We apologize for only being able to provide you with an explanation. Traditional speed and excitation control are still widely used control strategies in hydropower stations. In this study, we selected three control strategies for comparison by adjusting the damping factor value and modifying the target value setting. The correlation between the velocity of passing through the vibration zone and the pipeline pressure change was explored to meet the purpose of this study. Text explanations have been added (Line 345-348), and Fig. 8. has been modified. We have thought about it for a long time, but currently do not know how to compare the simulation results with more new control strategies.

6) The limitations and complexity of the proposed approach should discussed.

We sincerely appreciate the valuable comments. We have re-written this part (Line 374-379) according to the Reviewer’s suggestion. We discuss the advantages and limitations of this present study and give the need for the future scope.

We tried our best to improve the manuscript and made some changes in revised paper which will not influence the content and framework of the paper. We appreciate for Reviewers’ warm work earnestly, and hope the correction will meet with approval. Once again, thank you very much for your comments and suggestions.

Reviewer 3 Report

The authors are expected to address each and every comment raised at relevant sections and respective headings and revise their manuscript comprehensively.

Abstract: In this part, author should mention the practical application of the present study in the hydropower system. The numerical value should be provided to witness the observation and results.

Introduction:

Line 36- Please remove scholars and provide literatures.

Line 38- The reason for pressure fluctuation of draft tube (previous literatures) should be given for clear understanding of reader.  

Line 93 and 94: These lines should be improved and application of this study should be mentioned.

Problem statement

Line 97- Please provide the evidence of the statement.

Line 124- Please refer this sentence.

Line 152- approximately valid is not an accurate statement. The value should be provided and error should be estimated in order to find the solution.

Hamiltonian model

Line 168- addition of excitation to the PCH model of HU follows the assumptions. Please mention that for the understanding of the reader.

Line 183. Since this model consists of hydraulic condition with a penstock and a non elastic water column, whether it includes head losses. Please mention that in the paper.

Control Design

Line 210- Why it is complicated for high order matrix. Please explain in details.

Control structure

Please mention in the manuscript about the error in developing the model as well as in simulation. Without error calculation it is difficult to rely on the purity of the developed model.

Simulation Research

In Fig.2, only one load curve is given for the hydro-turbine and generator for different frequency. Why?

Fig.8, After 30s, it is difficult to differentiate between the case 1, 2 and 3. Please explain in the revised manuscript.

Conclusions

In conclusion section authors have discussed their observations. But they should discuss the advantages and limitations of this present study or when it is applicable (domain and range). What is the need of the present study and future scope?

NOTE: The authors must give technically strong answers to each of the above queries. Answers like "Addressed; Check the inclusion" will not be accepted.

Author Response

Dear Reviewers:

We feel great thanks for your professional review work on our article. As you are concerned, there are several problems that need to be addressed. According to your nice suggestions, we have made extensive corrections to our previous draft, the detailed corrections are listed below. The reviewer comments are laid out below in italicized font and specific concerns have been numbered. Our response is given in normal font and changes/additions to the manuscript are shown in the blue text.

Abstract:

C0. In this part, author should mention the practical application of the present study in the hydropower system. The numerical value should be provided to witness the observation and results.

Thank you for your reminder. We have re-written this part(Line 22-26) according to the Reviewer’s suggestion. Provide a numerical quantification of the simulation results.

Introduction:

C1. Line 36- Please remove scholars and provide literatures.

We sincerely thank the reviewer for careful reading. As suggested by the reviewer, we have corrected the “scholars” into “research”(Line 38). And the reference citation [2,3] has been adjusted.

C2. Line 38- The reason for pressure fluctuation of draft tube (previous literatures) should be given for clear understanding of reader.  

We have re-written this part according to the Reviewer’s suggestion (Line 37-39), and references [4,5] have been provided to explain the generation of pressure fluctuations.

C3. Line 93 and 94: These lines should be improved and application of this study should be mentioned.

Thanks for your suggestion. As suggested by the reviewer, we have added this part about application of this study (Line 96-100).

Problem statement

C4. Line 97- Please provide the evidence of the statement.

We sincerely appreciate the valuable comments. We have added a reference citation [18], which expresses it by sine or cosine that the amplitude and frequency of pressure fluctuation (Line 103).

C5. Line 124- Please refer this sentence.

We have added a reference citation [32], the vibration mechanics textbook (Line 129).

C6. Line 152- approximately valid is not an accurate statement. The value should be provided and error should be estimated in order to find the solution.

We have changed the description of ‘β(x) in Eq. (10) replacing β(x*) in a neighborhood around the equilibrium point is valid’ (Line 156-157). We are using the equilibrium point for variable constantization.

Hamiltonian model

C7. Line 168- addition of excitation to the PCH model of HU follows the assumptions. Please mention that for the understanding of the reader.

As suggested by the reviewer, we have given a hypothesis(Line 171-173) that the power oscillation approximates the Hamiltonian system input as an external excitation.

C8. Line 183. Since this model consists of hydraulic condition with a penstock and a non elastic water column, whether it includes head losses. Please mention that in the paper.

This article concerns the inclusion of pipeline head losses in the hydraulic turbine model, and an explanation has been added.(Line 187).

Control Design

C9. Line 210- Why it is complicated for high order matrix. Please explain in details.

We apologize for any questions you may have. The Hamiltonian model based on Lyapunov function as an energy function needs to ensure that its Hessian matrix satisfies the positive definite condition at the equilibrium point, so that the system with the damping control can be asymptotically stable. If the 8th-order Hamiltonian model is adopted in this paper, it will be more difficult to calculate the limited range of damping factors that satisfy the positive definite condition, and the optimal value cannot be given. In the text, we provided a brief explanation(Line 212-216) as it was not directly related to the content of this section, so we did not provide detailed elaboration.

Control structure

C10. Please mention in the manuscript about the error in developing the model as well as in simulation. Without error calculation it is difficult to rely on the purity of the developed model.

In this section, the Hamiltonian additional damping control is introduced into the system as an additional input signal of traditional speed regulation and excitation. Using the structure diagram, we only give the port information of the existing control system and the scheme of additional signal access. We does not involve the model development or graphical programming. Therefore, no error calculation is given. We think this is an omission we made. So we added an explanation (Line 237-238) for this figure.

Simulation Research

C11. In Fig.2, only one load curve is given for the hydro-turbine and generator for different frequency. Why?

We apologize for any confusion we may have caused. The article discusses the issue of active oscillation induced by pressure pulsation in the draft tube of the hydraulic turbine. As stated in Section 1, based on the external performance characteristics of pressure pulsation, we give the expression the mechanical power with pressure pulsation and obtain the equation of motion for active power oscillation. The power curve at different frequencies is presented here to demonstrate that when the frequency of pressure pulsation approaches the natural frequency of active power oscillation, resonance occurs and the amplitude of the power oscillation increases significantly. Therefore, it is only necessary to provide the power curve at different frequencies without the need to present changes in other system variables.

We apologize for only being able to provide you with an explanation. We have thought about it for a long time, but currently do not know how to add or modify the relevant description.

C12. Fig.8, After 30s, it is difficult to differentiate between the case 1, 2 and 3. Please explain in the revised manuscript.

Thanks for your suggestion. This is our omission. Text explanations have been added (Line 345-348) and Fig. 8. has been modified. In this section, we only investigate the power response curve of different control strategies when passing through the part-load vibration zone during the start-up. After 30 seconds, the unit leaves the vibration zone and gradually enters a steady state, and the influence of pressure pulsation disappears. Therefore, the response curve after 30s has not been specially marked. If necessary, the curves after 30s in Figure 8 can be discarded and not displayed.

Conclusions

 C13. In conclusion section authors have discussed their observations. But they should discuss the advantages and limitations of this present study or when it is applicable (domain and range). What is the need of the present study and future scope?

We sincerely appreciate the valuable comments. We have re-written this part (Line 374-379) according to the Reviewer’s suggestion.

NOTE: The authors must give technically strong answers to each of the above queries. Answers like "Addressed; Check the inclusion" will not be accepted.

We tried our best to improve the manuscript and made some changes in revised paper which will not influence the content and framework of the paper. We appreciate for Reviewers’ warm work earnestly, and hope the correction will meet with approval. Once again, thank you very much for your comments and suggestions.

Round 2

Reviewer 3 Report

The authors have revised the manuscript and it can now be considered.

Author Response

Dear Reviewers,

Thank you very much for your time involved in reviewing the manuscript and your very encouraging comments on the merits.

We would like to take this opportunity to thank you for all your time involved and this great opportunity for us to improve the manuscript. We hope you will find this revised version satisfactory.

Sincerely,

The Authors
